# Forest–Fruticulture Conversion Alters Soil Traits and Soil Organic Matter Compartments

**DOI:** 10.3390/plants11212917

**Published:** 2022-10-29

**Authors:** Bruna Firmino Enck, Milton Cesar Costa Campos, Marcos Gervásio Pereira, Fernando Gomes de Souza, Otavio Augusto Queiroz Santos, Yan Vidal de Figueiredo Gomes Diniz, Thalita Silva Martins, José Mauricio Cunha, Alan Ferreira Leite de Lima, Tancredo Augusto Feitosa de Souza

**Affiliations:** 1Postgraduate Program in Tropical Agronomy, Institute of Education, Agriculture and Environment, Federal University of Amazonas, Manaus 69067-005, Brazil; 2Postgraduate Program in Soil Science, Center of Agrarian Sciences, Federal University of Paraiba, Cidade Universitaria 58051-900, Brazil; 3Postgraduate Program in Soil Science, Institute of Agronomy, Federal University Rural of Rio de Janeiro, Seropédica 21941-901, Brazil; 4Agrotechnical School, Murupu Campus, Federal University of Roraima, Boa Vista 69300-000, Brazil; 5Postgraduate Program in Environmental Science, Institute of Education, Agriculture and Environment, Federal University of Amazonas, Manaus 69067-005, Brazil

**Keywords:** Amazon Tropical Rainforest, *Bixa orellana*, *Paullinia cupana*, *Theobroma grandiflorum*, Tropical Ultisols, tropical soil functioning

## Abstract

Fruticulture in the Amazonian Rainforest is one of the main causes of deforestation, biodiversity loss, and soil erosion. Fruticulture plays a key role in the soil traits and soil organic matter (SOM) compartments by altering the soil ecosystem. Our aim was to assess the influence of Forest–Fruticulture conversion on soil traits, and SOM fractions in Brazil’s Legal Amazon. The experiment was carried out in field conditions using four land uses as main treatments: *Bixa orellana*, *Theobroma grandiflorum*, *Paullinia cupana*, and the Amazon Rainforest. The soil physicochemical traits were analyzed using samples that were collected from 0–5, 5–10, and 10–20 cm soil depth by using grids (10 × 10 m) with 36 sampling points. Our results showed that the Fruticulture promoted an increase in bulk density, GMD, aggregate diameter, soil porosity, gravimetric moisture, sand, clay, carbon associated with humic acid, and, the sum of bases (K^+^, Ca^2+^, and Mg^2+^), while the Amazon Rainforest showed the highest values of silt, soil P content, SOC, p-SOC, m-SOC, carbon associated with fulvic acid, humine, and soil C stock. Overall, the fruticulture farming systems have negative effects on SOM compartments. The results of our study highlight the importance of considering fruticulture with endemic plant species by promoting soil fertility and soil aggregation.

## 1. Introduction

Brazil’s Legal Amazon (BLA) comprises the largest rainforest in the world. It covers an area of 5,016,136.3 km^2^ (59% of the Brazilian territory) where fruticulture is dominant when compared with other land uses (e.g., pasture, agriculture, etc.) [1]. In BLA, the influence of *Bixa orellana*, *Theobroma grandiflorum*, and *Paullinia cupana* on soil traits remains unclear. These fruticulture systems are used for annatto extraction, for production of cupuaçu fruit, and dietary supplement purposes, respectively [2]. The Amazon Rainforest has been recognized as one of the major sources of biodiversity. The soil traits from this ecosystem were previously described in Peralta et al. [3], and Marca-Zevallos et al. [4], and accordingly to these authors, the BLA provides favorable conditions for many endemic plant species to inhabit. However, its soil ecosystem suffers frequent soil fertility decline as influenced by land use. In this context, the relationship between soil biochemical properties and soil organic matter compartments is often restricted by land uses and anthropogenic activities [5]. 

The Tropical Ultisols from BLA are recognized with natural low fertility. This soil type is also very dependent on soil organic carbon compartments [6]. In this context, the soil’s physical and chemical properties are the main factors that we can use to assess changes in soil traits [7]. Land uses that are based on habitat simplification might play a significant role in the soil ecosystem, where there is a reduction in litter deposition, litter decomposition, and soil organic matter compartments as Banerjee and van der Heijden [8] and Cecagno et al. [9] have previously reported in their studies. Land covers that provide high litter deposition, have a positive influence on soil biochemical properties [10], and understanding the effect of fruticulture based on habitat simplification, which may regulate a wide range of ecosystem services into BLA, is essential to explain why soil fertility has a high and fast decline in such conditions, and why there is a high C loss [11].

In this work, we hypothesized that fruticulture imposes negative effects on the soil’s physical-chemical properties, and soil organic matter compartments [12]. It has been reported that plant species with high litter deposition and quality over a temporal scale may improve soil fertility by promoting some physical-chemical properties, such as soil organic carbon, nutrient content (P, Ca^2+^, and K^+^), and the chemical fraction of organic matter (Fulvic and humic acids) [13,14]. We also hypothesized that soil traits will be associated with certain plant patterns (e.g., plant community composition, nutrient export by shoots and fruits, root density, etc.) and, therefore, the variability among plots on soil traits will be a function of the cumulative effects of fruticulture as land use [15]. In tropical conditions, land use with high plant diversity is crucial to promoting soil traits, and soil organic matter compartments. Fruticulture that is based on monodominance in such conditions may negatively influence soil organic matter compartments by reducing litter deposition, and some geochemical pathways [1] promoting soil fertility and soil organic matter declines [11].

Our aim was to determine whether forest–fruticulture conversion alters the soil’s physical-chemical traits, and soil organic matter compartments in a tropical ecosystem of Brazil’s Legal Amazon. Based on the study developed by Souza et al. [1] and Brito et al. [2] we expected to find variations in soil physical-chemical traits, soil organic carbon, and soil organic matter compartments (by physical and chemical fractioning) as influenced by land use, especially in the ones with low litter deposition. To our knowledge, this is the first study assessing a complex database using soil physical-chemical traits and soil organic matter compartments on Brazil’s Legal Amazon To accomplish this, we performed a field study using grids and sampled soil in four land uses that are widely found in Brazil’s Legal Amazon with significant socioeconomic and environmental impact.

## 2. Results

Explanatory analysis of variance was performed to explore all data variability among the four studied land uses, soil depth, plots on soil traits, and soil organic matter compartments. In this explanatory analysis, we have considered land uses (df = 3), soil depth (df = 2), plots (df = 3), and their interaction, respectively, as sources of variation in a three-way ANOVA. Data from sampling points were nested by a plot. We did not find any significant differences in soil depth or the interactive effect of land uses and soil depth on soil traits. Thus, we performed a one-way ANOVA considering land use as a source of variation, and the observed results are shown in the “Influence of land use on soil traits in a tropical Ultisol from Brazil’s Legal Amazon” subsection. For soil organic matter compartments (SOC, p-SOC, m-SOC, C-Fulvic acid, C-Humic acid, and C-humin), we performed a two-way ANOVA, since we found significative differences between land use and soil depth interaction, and the observed results are shown in the “Influence of land use and soil depth on soil organic matter compartments in a tropical Ultisol from Brazil’s Legal Amazon” subsection.

### 2.1. Influence of Land Use on Soil Traits in a Tropical Ultisol from Brazil’s Legal Amazon

The results from the one-way ANOVA (*F*-test) showed significant differences among land use on all studied soil traits, but we did not find any significant difference among land use on soil pH. The highest values of bulk density, average diameter < 1.00 mm, soil microporosity, soil total porosity, MG, K^+^, Ca^2+^, Mg^2+^, SB, and V were found on plots where *P. cupana* was cultivated. For average diameter between 1.00 and 2.00 mm, average diameter < 1.00 mm, soil microporosity, clay content, Al^3+^, H^+^ + Al^3+^, and C.E.C the highest significant values were found on plots where *T. grandiflorum* was cultivated. Next, the highest significant values of GMD, WAD, average diameter > 2.00 mm, soil total porosity, MG, sand content, Al^3+^, and C.E.C. were found on plots where *B. orellana* was cultivated. Finally, the highest significant values of WAD, average diameter > 2.00 mm, silt content, Al^3+^, P, C.E.C., m, and CS were found in the Amazon Rainforest (Table 1).

### 2.2. Influence of Land Use and Soil Depth on Soil Organic Matter Compartments in a Tropical Ultisol from Brazil’s Legal Amazon

The results from the two-way ANOVA (*F*-test) showed significant differences among land use and soil depth on soil organic carbon (*p* < 0.001), particulate soil organic carbon (*p* < 0.01), soil organic carbon associated with minerals (*p* < 0.001), carbon associated with fulvic acid (*p* < 0.05), humic acid (*p* < 0.05), and humine (*p* < 0.05). The highest values of soil organic carbon, particulate soil organic carbon, soil organic carbon associated with minerals, and carbon associated with humine were found on plots within the Amazon Rainforest at 0.00–0.05 and 0.05–0.10 m. (We did not find any significant differences between these two layers). Next, the highest significant values of carbon associated with fulvic acid were found on plots where *T. grandiflorum* was at 0.05–0.10 and on plots within the Amazon Rainforest at 0.00–0.05 m. Finally, the highest significant values of carbon associated with humic acid were found on plots where *T. grandiflorum* was cultivated at 0.00–0.05 m (Table 2).

### 2.3. Multivariate Analysis

The PCA analyses showed that C associated with humine, C associated with minerals, soil organic carbon, average diameter between 1.00 and 2.00 mm, average diameter < 1.00 mm, WAD, and GMD were the main factors contributing to 65.56% of the data variance at a soil depth of 0 to 5 cm (Figure 1a). Next, C associated with fulvic acid, C associated with minerals, soil C stock, average diameter between 1.00 and 2.00 mm, average diameter < 1.00 mm, average diameter > 2.00 mm, WAD, and GMD were the main factors contributing to 70.65% of the data variance at a soil depth of 5 to 10 cm (Figure 1b). Finally, our PCA analyses showed that soil organic carbon, carbon associated with minerals, soil C stock, average diameter between 1.00 and 2.00 mm, average diameter < 1.00 mm, average diameter > 2.00 mm, WAD, and GMD were the main factors contributing to 75.16% of the data variance at a soil depth of 10 to 20 cm (Figure 1c). The analysis reinforced the dissimilarities among the land uses irrespectively to the selected soil depth on soil traits, and soil organic matter compartments (Figure 1). We found that: (i) C associated with humine has a significative contribution to data variance at a soil depth of 0–5 cm, and at this layer, the plots where *T. grandiflorum* was cultivated were better defined by aggregates with an average diameter < 1.00 mm; (ii) the plots where *B. orellana* was cultivated were better defined by WAD, GMD, aggregates with average diameter > 2.00 mm, Ca^2+^ and Mg^2+^ at a soil depth of 5–10 cm; (iii) the plots within the Amazon Rainforest were better defined by Ca^2+^, and Mg^2+^ at a soil depth of 10–20 cm; and (iv) the plots were *P. cupana* was cultivated were better defined by SOC, soil C stock, C associated with minerals at a soil depth of 10–20 cm.

## 3. Discussion

Our results emphasized the effects of land use on soil traits and soil organic matter compartments. Essentially, we wanted to understand how fruticulture following a conventional farming system can change the soil’s physical-chemical properties and soil organic matter compartments by promoting negative changes on the soil ecosystem using a field experiment with a natural ecosystem (the Amazon Rainforest as a reference area). In fact, land use type transformation affects soil properties through four main pathways, as described by Souza et al. [1]: (i) by altering habitat and energy provision with negative impacts on soil organic matter and litter compartments; (ii) by reducing plant diveristy and thus reducing litter deposition and quality; (iii) by altering the relationship among plants, soil, and other organisms; and (iv) by altering soil organic matter decomposition rate on the soil surface. In all fruticulture systems, we found increases in bulk density, GMD, aggregates classes (1.00–2.00 mm, and < 1.00 mm), soil porosity (micro, microporosity, and total), MG, sand, clay, the sum of bases (K^+^, Ca^2+^, and Mg^2+^), and C associated with humic acid [1,16]. On the other hand, all fruticulture systems decreased soil P content and soil organic matter compartments. These changes in soil physicochemical properties may disrupt the soil ecosystem by altering litter deposition, soil structure, water, and nutrient flow on the soil surface and root system [17,18]. 

Our hypothesis about the negative effects of fruticulture on soil traits and soil organic matter compartments was supported. Thus, considering the territory within Brazil’s Legal Amazon, there is scientific evidence proving the importance of land cover, and its litter as key factors for nutrient cycling, soil organic matter, and soil conservation in such conditions [2,19]. However, land uses based on conventional farming systems in tropical ecosystems can cause rapid land degradation (by decreasing soil organic carbon, and enhancing nutrient loss by run-off), which over time reduces some ecosystem services [16]. Soil aggregation is also strongly influenced by land uses in tropical rainforests as described by Six et al. [20] and Souza et al. [1].

In the context of Brazil’s Legal Amazon, the soil organic matter compartments (Soil C stock, SOC, p-SOC, m-SOC, C associated with fulvic acid, and humine) are important by promoting ecosystem services, such as soil organic matter sequestration and nutrient cycling [21]. The results of this study revealed that there were significant differences among the land covers on soil organic matter compartments. On one hand, the highest values of soil C stock, soil organic carbon, p-SOC, m-SOC, C associated with fulvic acid, and humine, were quantified on plots within the Amazon Rainforest. It could be considered that the natural ecosystem is crucial to promote soil traits and soil organic matter compartments. On the other hand, by considering the soil layers, the highest values of p-SOC were found in the topsoil due to the high deposition rate of litter and their decomposition. Thus, the maintenance of p-SOC content depends on the litter input in a way that benefits the soil organic matter mineralization [22], and this SOM fraction is more influenced by land uses than SOC [23]. Thus, the p-SOC can be used as a good indicator of soil quality for short-term land uses [24].

The Amazon Rainforest, through its characteristics, promotes soil quality, and the observed soil C stock values are in accordance with the results described by Frozzi et al. [25] and Jordão et al. [26]. These authors reported that the Amazon Rainforest presents a higher soil C stock on the soil’s surface when compared with monocropping systems within Amazon biomes. The high soil C stock in the natural ecosystem is related to the high m-SOC content. Souza et al. [19] have reported that soil C stock in tropical soils is composed of 80% of m-SOC. Other studies have reported that p-SOC (labile) and m-SOC (stable) are key factors to describe negative changes in soil ecosystems as influenced by land use. Several studies have also reported a positive correlation among soil C stock, p-SOC, and m-SOC [27]. The balance between these three SOM fractions is extremely important for carbon sequestration in tropical soils [28].

These changes in soil traits induced by land use in Amazon biomes are well reported [29]. In soil from Brazil’s Legal Amazon, the soil organic matter compartments are modulated by plant community composition. According to Souza et al. [1,15] plant community with the following characteristics: (i) high diversity; (ii) with nitrogen fixers; (iii) high soil reaction in the rhizosphere by the extrusion of H^+^; and (iv) high biomass production may enhance resource availability and improve both soil traits and soil organic matter compartments. However, in the fruticulture systems as land use, we lost all of these four characteristics. Scientific studies have highlighted the importance to consider the negative effects of conventional farming systems based on monocultures (i.e., *B. orellana*, *T. grandiflorum*, and *P. cupana*, with low plant diversity since they are based on monodominances) that may lead to less litter deposition and nutrient cycling [30]. 

Our hypothesis is that soil traits will follow certain plant patterns (e.g., plant community composition, nutrient export by shoots and fruits, root density, etc.) and, therefore, the variability among plots on soil traits will be a function of the cumulative effects of fruticulture as land use was supported for all fruticulture systems, which presented high values of the sum of bases (K^+^, Ca^2+^, and Mg^2+^) and low values of P content. The soil fertility in such conditions was influenced by management practices that created different pathways into the soil ecosystem, as we had found in our PCA analysis. It agrees with the investigations performed by Six et al. [20] and Souza et al. [15] that reported a positive correlation between soil organic matter compartments and soil P content in natural ecosystems, and positive influences on soil fertility as affected by soil management, as we observed in this study for the Amazon Rainforest, and all fruticulture systems, respectively [26].

For C associated with the chemical fractions of organic matter (Fulvic acid, humic acid, and humine), the Primary Amazon Rainforest showed higher values of C associated with fulvic acid and humine among the other studied fruticulture systems. In fact, the content of humic substances tends to reduce as affected by forest-to-agriculture conversion due to soil organic matter oxidation inside soil aggregates [31]. It can be supported by our results for GMD, SOC, aggregate classes, and C associated with humic substances in the Amazon Rainforest. Although our experiment was not designed to directly test whether litter and root traits affect soil organic matter compartments, we must consider that the highest plant diversity can be positively correlated with soil organic matter compartments. Considering our results, we propose some potential correlations (mechanisms) for land use-mediated C fractions, as we summarized in our PCA analysis. The first hypothesis was partially supported since fruticulture promoted SOC, soil C stock, and C associated with minerals, which was the opposite of our hypothesis; however, in the natural ecosystem, we found the highest values of C associated with fulvic acid and humine, as we expected [32]. Our second hypothesis was supported by the results that the higher C stock showed a high SOC but with lower C associated with fulvic acid and humine, as we expected. However, the second hypothesis was challenged by a greater litter deposition in the fruticulture system used for land use [19]. Furthermore, we did not find any statistically significant interaction between land use and soil depth for soil traits, suggesting a consistent influence of land use on soil physical-chemical properties across soil profiles [21]. 

Land use, soil depth, soil traits (physical and chemical), and soil organic matter compartments were five important drivers of the entire soil function (Souza et al., 2020). In fact, the natural ecosystem showed higher values of SOC stock, p-SOC, and m-SOC when compared with the other studied fruticulture farming systems. According to the investigation performed by Oliveira et al. [33] the SOC stock varies as influenced by plant community composition and dissimilar biomes. This study highlights the importance of land cover and root systems as the main source of SOC in tropical agroecosystems. Tropical soils located in the Amazon Rainforest have been described with high SOC stock, considering their traits to accumulate SOC in deep layer trough soil profile. In general, the organic matter compartments of these soils are influenced by their physical traits, soil type, and slope [30]. For the p-SOC and m-SOC results, we expected to find the lowest values of p-SOC in the fruticulture farming systems and the highest values of m-SOC in the natural ecosystem These fractions are considered an indicator of soil management and recalcitrant fraction, respectively [32]. 

Considering the abundance of humic substances, the C-humine was the most abundant fraction in all studied land uses. According to the investigation performed by Marinho Jr et al. [24] it is a common phenomenon in tropical soils due to: (i) the high development level of humine and its resistance to microbial activity (decomposition); and (ii) the interaction between the humine fraction and soil mineral fraction. These authors also described that the highest C-humine content could be correlated with the size of soil particles and high soil stability. Our study also revealed that the highest values of C-Fulvic acid in the natural ecosystem are related to its natural soil acidity, which creates an unfavorable condition for microbial activity and decreases soil fertility by reducing soil organic matter decomposition [34]. In this instance, we must consider the results of C-Fulvic acid as indicators for alternative farming systems that consider soil fertility and SOC maintenance. For C-humic acid, our study revealed that *T. grandiflorum* was the only studied land use that showed the highest values of this fraction. It indicated a better soil quality when compared with the other studied land uses, since this SOM fraction is considered the most stable fraction that favors the CEC on soil superficial layers [22,35].

## 4. Materials and Methods

### 4.1. Study Site, Climatic Conditions, and Soil Type

The study site was carried out in long-term field experiments using three crop systems and a natural ecosystem (as reference area) at the Amazon Lowlands basin located in Canutama, Amazonas, Brazil (08°11′22″ S, 64°00′83″ W). The climate in the experiment area is Am-type following Köppen climate classification (e.g., tropical monsoon climate with a short dry period), with average annual precipitation and air temperature of 2950 mm and 26 °C, respectively [36]. Soil type is associated with recent and ancient alluvial sediments from the Quaternary period, and it is characterized by the presence of large tabular well-defined reliefs, containing smooth slopes and poor natural drainage [37]. The soil type in the study site was classified as Ultisol [38].

### 4.2. The Studied Crop Systems and Natural Ecosystem

The crop systems were characterized by conventional monocropping systems with endemic perennial tree species: (i) *Bixa orellana* L. (Commonly known as “Annatto”) in a 3-year monocropping system (MS), 500 plants ha^−1^, with a litter deposition of 3.95 T ha^−1^ year^−1^; (ii) *Theobroma grandiflorum* (Willd. ex Spreng.) K. Schum. (Commonly known as “Cupuaçu”) in a 7-year MS, 500 plants ha^−1^, with a litter deposition of 4.44 T ha^−1^ year^−1^; and; (iii) *Paullinia cupana* Kunth, (Commonly known as “Guaraná”) in a 7-year MS, 400 plants ha^−1^, with a litter deposition of 5.63 T ha^−1^ year^−1^. These three monocropping systems were selected because they used perennial plant species that are classified as endemic from Brazil’s Legal Amazon, Amazonas, Brazil. In all crop systems, the forest–fruticulture conversion was carried out after timber and the use of fire to clear the natural forest. No liming, fertilizers, or herbicides were used in these three crop systems. They are native to the Amazon Forest and they are widely used by the indigenous community from the Amazon Lowlands basin. The natural ecosystem is inserted inside the National Forest of Balata-Tufari. It was classified as Amazon rainforest (Moist Broadleaf Tropical Rainforest composed of a mix of plants from Annonaceae, Apocynaceae, Arecaceae, Bromeliaceae, Clusiaceae, Fabaceae, Lecythidaceae, Malvaceae, Meliaceae, Myrtaceae, Olacaceae, Orchidaceae, and Rubiaceae) that is monitored for conservation purposes since the 2000s. This ecosystem had a litter deposition of 8.38 T ha^−1^ year^−1^.

### 4.3. Experimental Design

We analyzed the effects of different monocropping systems on soil traits and soil organic matter compartments and compared their results with the ones from a natural ecosystem that was used as a reference area. Before sampling soil for analyses (physicochemical characterization, and soil organic matter fractions), the following criteria were established: (a) each studied condition had 4 permanent plots (60 × 60 m) with homogenous plant stand; (b) into each plot, we sampled 36 points in gridded areas (10 × 10 m); (c) each sample included undisturbed and disturbed samples at the following depths: 0–5 cm, 5–10 cm, and 10–20 cm. More details about the experimental area are given in Figure 2.

### 4.4. Soil Sampling

We collected 432 undisturbed and 432 disturbed soil samples for soil physical and chemical characterization, respectively. For soil physical characterization, we determined aggregate traits, such as weighted average diameter (WAD), geometric mean diameter (GMD), and aggregate classes. Soil bulk density was measured by considering the weight of soil per unit volume of a metallic cylinder [39]. Soil macroporosity (S-MaP), soil microporosity (S-MiP), and total soil porosity (S-TP) were determined as described by Teixeira et al. [40]. Soil texture was determined by a particle size analysis of the dispersed soil [41]. The clay was flocculated by adding NaCl solution as a chemical dispersing agent [39]. For chemical characterization, soil reaction (pH) was measured in a suspension of soil and distilled water (1:2.5 v:v, soil:water suspension) [39]. The potassium chloride extraction method was used to determine exchangeable Al^3+^ [42]. All soil exchangeable cations (K^+^, Ca^2+^, Mg^2+^) were determined by an extraction method using atomic absorption spectrophotometer for Ca^2+^ and Mg^2+^ and a flame photometer for K^+^ [41]. Available phosphorus (Olsen’s P) was determined colorimetrically using a spectrophotometer at 882 nm by extraction with sodium bicarbonate for 30 min [43]. Cation exchange capacity was measured using the following equation: C.E.C. (cmolc kg^−1^) = K^+^ + Ca^2+^ + Mg^2+^ + H^+^ + Al^3+^ [39].

### 4.5. SOM Fractioning

The samples were air-dried and passed through a 2-mm sieve. The total organic carbon was determined by the rapid dichromate oxidation method following the protocol described in Okalebo et al. [44]. Soil organic carbon stock (SOC stock) was determined as described by da Silva et al. [23]. The physical fractioning of the soil organic matter was carried out following the methodology described by Cambardella and Elliot [45]. We determined: (i) particulate soil organic carbon (p-SOC) by weighting the retaining material in a 0.05-mm sieve after 16h of homogenization in sodium hexametaphosphate solution; and (ii) soil organic carbon associated with minerals (m-SOC) by the difference between TOC content and p-SOC content. Soil organic matter chemical fractions (Fulvic acid, humic acid, and humin) were determined according to the International Humic Substances Society [46,47,48].

### 4.6. Statistical Analysis

All data were analyzed using R statistical software [49]. We used arcsine square root to transform all dependent variables to meet the assumption of normal distribution. We tested all variables for normality using the “Shapiro.test()” function. The “Moran.I function” from the “ape” package was used to find spatial autocorrelation among soil samples [15]. None of the studied variables were found to have any site effect (e.g., no spatial relationship) with the sampling point. Since the Shapiro–Wilk test indicated Gaussian distribution and our samples were spatially independent, it enabled us to use one-way ANOVA to compare crop systems. As we did not find interaction effects between crop systems and soil depth, we also used a one-way ANOVA to compare soil depths (0–5; 5–10; and 10–20 cm). We used Bonferroni’s test (“stats” package) to compare the variables at plots with the different crop systems and soil depths. We performed a principal component analysis (PCA) using the “vegan” package to outline the dissimilarities among the soil’s physicochemical traits, the soil organic matter fractions, and the studied crop systems [23].

## 5. Conclusions

The forest–fruticulture conversion determined soil traits and soil organic matter compartments in a tropical Ultisol from Brazil’s Legal Amazon. The fruticulture farming systems showed high bulk density, GMD, aggregates classes (1.00–2.00 mm, and <1.00 mm), soil porosity (micro, microporosity, and total), MG, sand, clay, the sum of bases (K^+^, Ca^2+^, and Mg^2+^), and C associated with humic acid under field conditions. Our findings suggest that fruticulture farming systems have negative effects on soil organic matter compartments by the loss of four main traits: (i) high diversity, (ii) abundance of nitrogen fixers, (iii) high soil reaction in the rhizosphere by the extrusion of H^+^; and (iv) high biomass production. The results of our study highlight the importance of considering fruticulture farming systems with endemic plant species from Brazil’s Legal Amazon as a soil conditioner (by promoting soil fertility, nutrient cycling, and soil aggregation). However, we also must consider their impacts on soil organic matter compartments. Thus, long-term experiments considering *B. orellana*, *T. grandiflorum*, and *P. cupana* may exploit positive feedback among soil traits, soil organic matter compartments, and soil health, since the results of C-Fulvic acid and C-Humic acid indicated a good level of soil quality in such systems.

## Figures and Tables

**Figure 1 plants-11-02917-f001:**
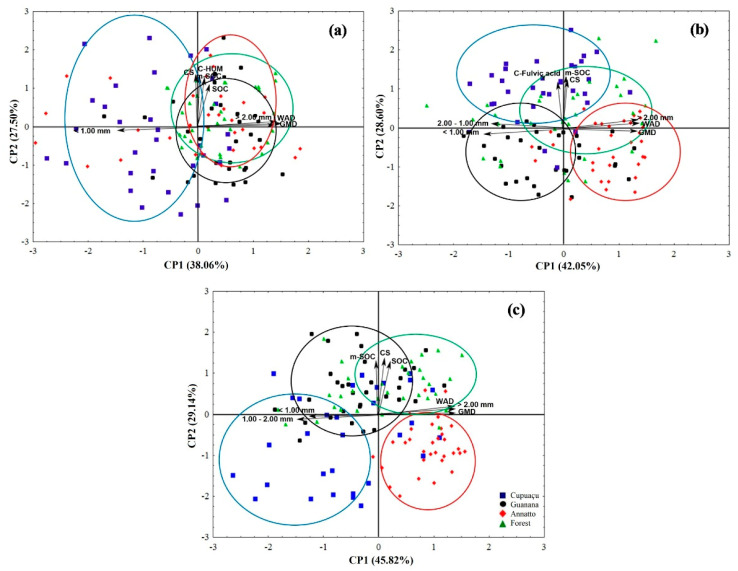
Principal component analysis (PCA) score plot for land use, soil traits, and soil organic matter compartments in the 0–5 (**a**), 5–10 (**b**), and 10–20 cm (**c**) as influenced by land use in Brazil’s Legal Amazon. Polygons represent land use. Two axes represent 65.56, 70.65, and 75.16% of the data variance at 0–5, 5–10, and 10–20 cm, respectively.

**Figure 2 plants-11-02917-f002:**
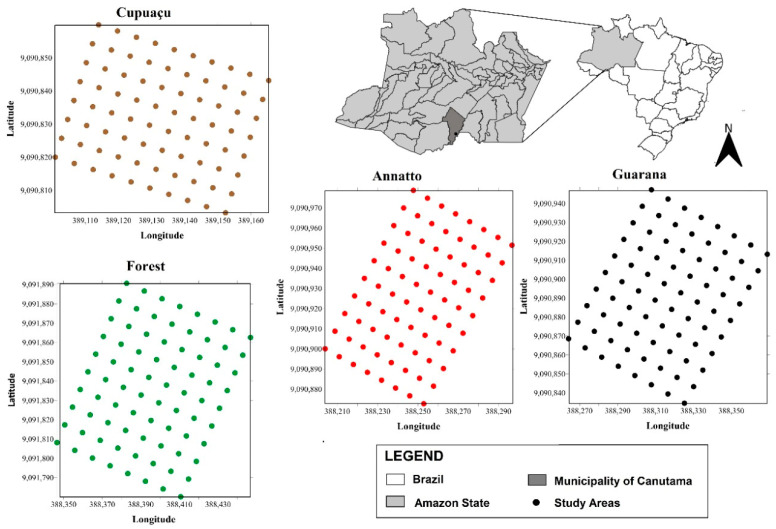
Location map of the study site and sampling points in the rainforest areas both native and under conversion to crop cultivation in southern Amazonas.

**Table 1 plants-11-02917-t001:** Soil traits in different land use from Brazil’s Legal Amazon.

Soil Traits	*P. cupana*	*T. grandiflorum*	*B. orellana*	Amazon Rainforest	*F*-Value
	Physical traits				
Bulk Density (g cm^−3^)	1.11 (0.01) a	1.07 (0.02) b	1.04 (0.02) b	0.92 (0.05) c	17.92 ***
Geometric mean diameter (mm)	2.49 (0.04) b	2.38 (0.01) c	2.72 (0.04) a	2.57 (0.02) b	13.44 ***
Weighted average diameter (mm)	2.96 (0.02) b	2.90 (0.05) b	3.12 (0.04) a	3.04 (0.02) a	9.47 **
>2.00 (mm)	86.43 (0.31) b	83.89 (0.27) b	92.37 (0.54) a	89.53 (0.38) a	9.98 **
1.00–2.00 (mm)	3.36 (0.11) b	4.33 (0.12) a	1.44 (0.09) d	2.52 (0.17) c	18.35 ***
<1.00 (mm)	10.27 (0.02) a	11.57 (0.06) a	6.18 (0.02) c	8.35 (0.03) b	15.91 ***
Soil macroporosity (mm^3^ mm^−3^)	9.86 (0.05) c	13.20 (0.12) a	12.48 (0.10) b	12.46 (0.01) b	13.79 ***
Soil microporosity (mm^3^ mm^−3^)	37.04 (0.21) a	19.68 (0.14) c	33.19 (0.25) b	20.79 (0.11) c	18.26 ***
Total soil porosity (mm^3^ mm^−3^)	47.41 (0.51) a	37.11 (0.47) b	45.16 (0.45) a	36.79 (0.61) b	9.31 **
MG (mm^3^ mm^−3^)	33.35 (0.40) a	27.51 (0.69) b	32.49 (1.09) a	27.59 (0.41) b	9.53 **
Sand (g kg^−1^)	365.94 (11.01) b	277.41 (9.01) c	384.64 (8.37) a	238.39 (2.11) d	12.34 ***
Silt (g kg^−1^)	405.31 (9.31) c	450.31 (12.51) b	386.74 (21.09) c	524.70 (17.01) a	14.58 ***
Clay (g kg^−1^)	223.91 (3.31) d	267.64 (2.38) a	239.77 (5.57) c	260.47 (3.67) b	11.94 ***
	Chemical traits				
Soil pH	3.89 (0.01) a	3.92 (0.02) a	3.87 (0.01) a	3.74 (0.02) a	3.45 ^ns^
Al^3+^ (cmol_c_ dm^−3^)	4.52 (0.01) b	5.11 (0.04) a	5.04 (0.04) a	5.04 (0.06) a	8.94 *
H^+^ + Al^3+^ (cmol_c_ dm^−3^)	9.34 (0.09) c	10.42 (0.08) a	9.96 (0.01) b	9.95 (0.02) b	9.31 **
K^+^ (cmol_c_ dm^−3^)	0.11 (0.01) a	0.06 (0.02) b	0.07 (0.01) b	0.08 (0.03) b	8.56 *
Ca^2+^ (cmol_c_ dm^−3^)	1.21 (0.03) a	0.86 (0.01) b	0.92 (0.03) b	0.71 (0.02) c	11.49 ***
Mg^2+^ (cmol_c_ dm^−3^)	0.42 (0.02) a	0.19 (0.01) c	0.26 (0.01) b	0.19 (0.02) c	10.59 **
P (mg dm^−3^)	6.04 (0.08) b	3.16 (0.11) d	4.87 (0.09) c	7.73 (0.10) a	15.24 ***
SB (cmol_c_ dm^−3^)	1.94 (0.01) a	1.14 (0.01) c	1.29 (0.02) b	1.04 (0.01) c	11.13 ***
C.E.C (cmol_c_ dm^−3^)	10.83 (0.02) b	11.62 (0.05) a	11.36 (0.03) a	11.14 (0.04) a	10.16 **
V (%)	16.40 (0.31) a	10.12 (0.15) c	11.71 (0.27) b	9.67 (0.23) c	11.09 ***
m (%)	71.75 (1.01) c	82.71 (0.68) a	79.75 (1.23) b	83.97 (0.89) a	11.59 ***
CS (t ha^−1^)	8.67 (0.28) c	9.18 (0.27) b	9.99 (0.13) b	11.44 (0.97) a	12.36 ***

Standard error in parentheses. ***, **, *, and ^ns^ represent significant effects by one-way ANOVA at *p* < 0.001, 0.01, 0.05, and not-significant, respectively. Within land use, the same small letters represent no significant differences by Bonferroni’s test (*p* < 0.05).

**Table 2 plants-11-02917-t002:** Soil organic matter compartments among land use and soil depth in a Ultisols from Brazil’s Legal Amazon.

Soil Depth (m)	*B. orellana*	*T. grandiflorum*	*P. cupana*	Amazon Rainforest
SOC (g kg^−1^)				
0.00–0.05	20.67 (0.93) Ac ^1^	22.31 (1.03) Ab	22.43 (1.21) Ab	26.02 (1.57) Aa
0.05–0.10	12.32 (0.81) Bc	17.20 (2.01) Bb	14.59 (0.84) Bc	21.28 (0.98) Aa
0.10–0.20	10.79 (0.56) Bb	6.68 (0.98) Cc	7.73 (0.34) Cc	12.86 (1.03) Ba
p-SOC (g kg^−1^)				
0.00–0.05	6.27 (0.12) Ac	7.14 (0.09) Ab	7.66 (0.09) Ab	8.53 (0.10) Aa
0.05–0.10	2.67 (0.23) Bb	2.80 (0.11) Bb	3.44 (0.11) Bb	7.27 (0.10) Aa
0.10–0.20	2.54 (0.17) Bb	1.30 (0.03) Bc	3.45 (0.06) Bb	4.18 (0.03) Ba
m-SOC (g kg^−1^)				
0.00–0.05	14.62 (0.13) Ab	13.45 (1.17) Ab	14.58 (0.93) Ab	16.18 (1.17) Aa
0.05–0.10	9.42 (0.15) Bb	14.12 (0.16) Aa	10.25 (0.81) Bb	13.47 (0.99) Ba
0.10–0.20	8.74 (0.21) B	5.23 (0.81) B	3.98 (0.27) C	8.94 (0.51) C
C-Fulvic acid (g kg^−1^)				
0.00–0.05	2.71 (0.05) Ab	2.95 (0.01) Ab	1.92 (0.01) Ac	4.53 (0.01) Aa
0.05–0.10	2.31 (0.03) Ab	3.31 (0.02) Aa	2.04 (0.05) Ab	2.71 (0.05) Bb
0.10–0.20	-	-	-	-
C-Humic acid (g kg^−1^)				
0.00–0.05	2.91 (0.03) Aa	3.50 (0.05) Aa	2.66 (0.02) Ab	2.98 (0.05) Ab
0.05–0.10	2.03 (0.01) Ba	2.86 (0.01) Ba	1.19 (0.03) Bb	1.48 (0.01) Bb
0.10–0.20	-	-	-	-
C-Humine (g kg^−1^)				
0.00–0.05	15.37 (0.01) Ab	15.73 (0.02) Ab	17.53 (0.02) Aa	17.41 (0.01) Aa
0.05–0.10	7.83 (0.03) Bc	11.12 (0.05) Bb	10.82 (0.05) Bb	16.86 (0.02) Aa
0.10–0.20	-	-	-	-

Standard error in parentheses. “-“ indicates trace values with no significant differences. Within land use and soil depths, ^1^ the same small and capital letters represent no significant differences by Bonferroni’s test (*p* < 0.05), respectively.

## Data Availability

Not applicable.

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
