# Peer review of "Forest–Fruticulture Conversion Alters Soil Traits and Soil Organic Matter Compartments"

_plants, 2022, doi:10.3390/plants11212917_

Round 1

Reviewer 1 Report

This manuscript reports on the impact of fruitculture on Amazon Forest via analysis of the physico-chemical properties of the collected soil samples. Overall, it is a well designed and presented study. The follows are a few minor comments.

1.       L30. Incorporate some conclusive summary for this study.

2.       L121. “2.1”.

3.       L170 vs L217. Revise figure numbers in this section (and others like L379).

4.       L218. Add “(c)”.

5.       L220. What are the percentage numbers in the figure?

6.       Figures. The resolution is very poor.

Author Response

Comments and Suggestions for Authors

Reviewer #1

This manuscript reports on the impact of fruitculture on Amazon Forest via analysis of the physico-chemical properties of the collected soil samples. Overall, it is a well designed and presented study. The follows are a few minor comments.

We are so glad do read your comments. Thanks for your feedback.

L30. Incorporate some conclusive summary for this study.

We understand the reviewer point-of-view. However, we must use at least 200 words accordingly do authors’ instructions. So, we have re-written some phrases to attend the reviewer suggestion. See L28-30.

L121. “2.1”.

We have adjusted it accordingly. See L101.

L170 vs L217. Revise figure numbers in this section (and others like L379).

Thanks. We have revised and adjusted all figure numbers. See L165, 168, 172, 174, 353.

L218. Add “(c)”.

Agreed. See L184.

L220. What are the percentage numbers in the figure?

The numbers are the contribution of PC1 and PC2 on data variance. See L185-186.

Figures. The resolution is very poor.

Thanks for this comment. We tried our best to improve the quality (resolution) of both figures (Fig. 1 and Fig.2)

Reviewer 2 Report

1. Three monocropping systems, please introduce their density, stand age, harvest and other factors affecting soil properties.

2. It is suggested to supplement the basic situation of the plot, such as geographical coordinates, growth, and the difference between the plot and the pre-conversion stand.

3. Soil microporosity, how are the relevant indicators calculated? how do they relate to bulk density?

4. Table 2, what dose “-” stand for?

5. In discussion, it is suggested to strengthen the analysis of why land use type transformation affects soil properties.

6. In discussion, do you want to analyze carbon sequestration function through organic carbon components?

7. P311-312, five important drivers, please check it.

8. Overall, did fruticulture farming systems have a positive or negative effect on soil traits?

Author Response

Reviewer #2

Three monocropping systems, please introduce their density, stand age, harvest and other factors affecting soil properties.

Agreed. Plant density and stand age were introduced accordingly. See L320-321; 322-323; and 324. Further, we have added information about factors affecting soil properties. See L328-329.

It is suggested to supplement the basic situation of the plot, such as geographical coordinates, growth, and the difference between the plot and the pre-conversion stand.

We have improved the Figure 2 to clarify all scientific information.

Soil microporosity, how are the relevant indicators calculated? how do they relate to bulk density?

We calculated soil porosity (micro-, macro-, and total porosity) as described by Teixeira et al. [40]. See L365-366. We expected to find a direct relationship with them. However, we cannot find it through the statistical analysis. Thus, considering that we did not find any significative relationship with porosity and bulk density, we did not discuss it into the discussion section.

Table 2, what dose “-” stand for?

It indicates trace values for humic substances with no significative differences. See L158.

In discussion, it is suggested to strengthen the analysis of why land use type transformation affects soil properties.

Agreed. See L193-199.

In discussion, do you want to analyze carbon sequestration function through organic carbon components?

It was not our aim with this study and considering our deadline to return this review we preferred to not include this statement in our manuscript.

P311-312, five important drivers, please check it.

That is correct. Because in soil traits we must consider the physical and chemical traits. We have included this statement in the text to be clear. See L284.

Overall, did fruticulture farming systems have a positive or negative effect on soil traits?

Overall, the fruticulture farming systems have negative effects on SOM compartments. See L28; L415-419.

Reviewer 3 Report

Overall, the paper by Enck et al. discusses the influence of Forest-Fruticulture conversion on soil traits, and soil organic matter (SOM) compartments in a tropical Ultisol from Brazil’s Legal Amazon. Three typical land-use crop systems and one natural ecosystem were selected as study area, and the soil physicochemical traits were analyzed using samples (disturbed and undisturbed soil samples). Meanwhile, the manuscript is well structured and organized. Therefore, I suggest it to be published in the Plants, with some minor suggestions for its approvement, as follows:

Abstract and keywords

The research significance should be added at the end of the abstract.

Line 31: the keywords “Theobroma grandiflorum” should use italics.

Introduction

Line 50: the references showed be corrected.

Results

Line 174, line 177, line 181, and line 183: change “Fig. 2” to “Fig. 1”.

Line 379: change “Fig. 1.” to “Fig. 2.”.

Materials and Methods

Line 377: This part emphasized each sample including undisturbed and disturbed, why not analysis the difference between undisturbed and disturbed in results and discussion? I thought it is also important for us to understand the soil traits, and soil organic matter influenced by anthropogenic activities.

In addition, the image of Fig. 1 and Fig. 2 are not very clear.

Author Response

Reviewer #3

Overall, the paper by Enck et al. discusses the influence of Forest-Fruticulture conversion on soil traits, and soil organic matter (SOM) compartments in a tropical Ultisol from Brazil’s Legal Amazon. Three typical land-use crop systems and one natural ecosystem were selected as study area, and the soil physicochemical traits were analyzed using samples (disturbed and undisturbed soil samples). Meanwhile, the manuscript is well structured and organized.

Thanks for your feedback, and suggestions.

Therefore, I suggest it to be published in the Plants, with some minor suggestions for its approvement, as follows:

Abstract and keywords

The research significance should be added at the end of the abstract.

Agreed. We have added a conclusive summary for this study at the end of the abstract. See L28-30.

Line 31: the keywords “Theobroma grandiflorum” should use italics.

We have adjusted it as recommended. See L30.

Introduction

Line 50: the references showed be corrected.

Agreed. See L50

Results

Line 174, line 177, line 181, and line 183: change “Fig. 2” to “Fig. 1”.

Thanks. We have revised and adjusted all figure numbers. See L165, 168, 172, 174

Line 379: change “Fig. 1.” to “Fig. 2.”.

Agreed. See L353

Materials and Methods

Line 377: This part emphasized each sample including undisturbed and disturbed, why not analysis the difference between undisturbed and disturbed in results and discussion? I thought it is also important for us to understand the soil traits, and soil organic matter influenced by anthropogenic activities.

We got the reviewer’s point-of-view. In fact, we discussed about the differences between physical traits (determined through the undisturbed samples), chemical and SOM compartments (determined through the disturbed samples).

In addition, the image of Fig. 1 and Fig. 2 are not very clear.

We tried our best to fix them.

Reviewer 4 Report

Enck et al reported how soil physico-chemical proerties and SOM fractions varies across 4 different land uses in Amazon rainforests. They found fruticulture can result in losses in soil C stock via decreased p-SOC and m-SOC compared to natural rainforest. I find the data and findings are of interest to publication, yet the language requires intensive improvement.

Introduction: first two paragraphs was written badly. Need improvement. 

L27 what is MG?

L38-40 Italics for Latin names.and carbon associated with humine on plots within Amazon Rainforest

L46 "relationship between soil ecosystem (soil biochemical properties, and soil organic matter compartments)" improper usage of "relationship", maybe "soil quality"

L48-L50 Unclear sentence

L51-52 grammar; L53 "assume"-> "assess"; L59 Fruticulture

L62 "promotes" -> "imposes"; define "soil traits"

L67 "follow" -> "will be associated with"

L92 "land uses depth"->interactive effect of land uses and soil depth

L107 "m"?

L158 "and carbon associated with humine were found on plots within Amazon Rainforest"; L160 same problem

L184 contribution "with" -> "to"

L236 no litter data to support this claim. At most you can claim the land covers affects the soil ecosystem functions. It is possible the fruticulture affects soil ecosystem via root-soil interaction.

L239 "improving" ->enhancing

L244 "decomposition" -> "sequestration"

L267 "promoted"->"induced"

These are not all the grammar error. I suggest authors seek further assistance for language.

Author Response

Enck et al reported how soil physico-chemical properties and SOM fractions varies across 4 different land uses in Amazon rainforests. They found fruticulture can result in losses in soil C stock via decreased p-SOC and m-SOC compared to natural rainforest. I find the data and findings are of interest to publication, yet the language requires intensive improvement.

Introduction: first two paragraphs was written badly. Need improvement.

We tried our best to improve both paragraphs (35-60).

L27 what is MG?

It is gravimetric moisture. See L25.

L38-40 Italics for Latin names.and carbon associated with humine on plots within Amazon Rainforest

Agreed. See L38-39.

L46 "relationship between soil ecosystem (soil biochemical properties, and soil organic matter compartments)" improper usage of "relationship", maybe "soil quality"

We have improved this statement accordingly. See L45-46.

L48-L50 Unclear sentence

We have excluded it to be clear and to avoid misunderstanding.

L51-52 grammar; L53 "assume"-> "assess"

Agreed. See L50.

L59 Fruticulture

Agreed. See L69.

L62 "promotes" -> "imposes"; define "soil traits"

Agreed. See L59-60.

L67 "follow" -> "will be associated with"

Agreed. See L64-65.

L92 "land uses depth"->interactive effect of land uses and soil depth

We have adjusted it accordingly. L90-91

L107 "m"?

We have excluded it. See L105.

L158 "and carbon associated with humine were found on plots within Amazon Rainforest"; L160 same problem

Agreed. See L146-147; L149.

L184 contribution "with" -> "to"

Agreed. See L174.

L236 no litter data to support this claim. At most you can claim the land covers affects the soil ecosystem functions. It is possible the fruticulture affects soil ecosystem via root-soil interaction.

Litter data was described in the characterization of each studied Fruticulture system. See L329, 331, and 333.

L239 "improving" ->enhancing

Agreed. See L213.

L244 "decomposition" -> "sequestration"

Agreed. See L218.

L267 "promoted"->"induced"

Agreed. See L241.

These are not all the grammar error. I suggest authors seek further assistance for language.

We tried our best to revise all content by following the reviewers’ comments and suggestions.

Reviewer 5 Report

The purpose of this study was to evaluate the effects of

20 forest-fruticulture conversion on soil characteristics and soil organic matter (SOM) fractions in Brazil's 21 Legal Amazon. Legal Amazon using four different land uses. In the study, the authors hypothesized that fruticulture promotes negative effects on soil traits traits and soil organic matter compartments They found that plant species

with high deposition and litter quality on a time scale can improve soil fertility. This hypothesis was tested using appropriately selected samples. The authors point out that there has not yet been such a study based on a comprehensive database using soil physico-chemical characteristics and organic matter compartments in Brazil. organic matter on the Brazilian Legal Amazon, To test the hypothesis, the authors used analysis of variance. This was a so-called Three way of ANOVA. The effect of the main factors and interactions between them were examined. a two-way ANOVA was used in a further step. and one-way ANOVA. Surprising to me is the result that there were no significant differences between land use on soil pH. The statistical methodology was supplemented with PCA. This is very good, because although ANOVA provides some information it is a basic method that needs to be supplemented. The discussion in this paper, for me, is not very coherent. I think it is also too long. The conclusions, on the other hand, I rate very well. I miss the addition of a causal explanation of the results. In simpler terms, why such results were obtained. I suggest supplementing the methodology with a CART analysis. 

Author Response

Comments and Suggestions for Authors

Reviewer #4

The purpose of this study was to evaluate the effects of forest-fruticulture conversion on soil characteristics and soil organic matter (SOM) fractions in Brazil's Legal Amazon. Legal Amazon using four different land uses.

That is correct. We evaluated the effects of forest-fruticulture conversion on soil traits (physical and chemical) and SOM compartments in BLA. Among the four different land uses, three were classified as fruticulture farming systems, and the last one was classified as natural ecosystem (Amazon Rainforest).

In the study, the authors hypothesized that fruticulture promotes negative effects on soil traits and soil organic matter compartments They found that plant species with high deposition and litter quality on a time scale can improve soil fertility.

That is correct. Actually, we found that the fruticulture farming systems have negative effects on SOM compartments, and we highlighted the importance of considering fruticulture with endemic plant species by promoting soil fertility and soil aggregation

This hypothesis was tested using appropriately selected samples.

Thanks for this comment. We tried our best to avoid pseudo-replicates and autocorrelation.

The authors point out that there has not yet been such a study based on a comprehensive database using soil physico-chemical characteristics and organic matter compartments in Brazil. organic matter on the Brazilian Legal Amazon.

That is correct.

To test the hypothesis, the authors used analysis of variance. This was a so-called Three way of ANOVA. The effect of the main factors and interactions between them were examined. a two-way ANOVA was used in a further step. and one-way ANOVA.

That is correct. We started with an explanatory ANOVA, and thus we tested the best ANOVA type that fitted well with our sources of variance.

Surprising to me is the result that there were no significant differences between land use on soil pH.

We expected to do not find differences on soil pH as affected by land uses, since limestone was not used in any of the studied fruticulture farming systems. We have added this information into the methodology. See L335.-336.

The statistical methodology was supplemented with PCA. This is very good, because although ANOVA provides some information it is a basic method that needs to be supplemented.

That is correct. The reviewer got our point-of-view when we have chosen to combine a univariate analysis (ANOVA) with a multivariate one (PCA).

The discussion in this paper, for me, is not very coherent. I think it is also too long.

We respect the reviewer’s opinion. However, we tried our best to write a discussion by considering our hypothesis, our aims, our main findings, and the causal explanation for each finding that we observed in our field study. All scientific statement were supported by update references with coherent relationships (especially that ones that were done into the Amazon Rainforest).

The conclusions, on the other hand, I rate very well.

We are glad to read this comment.

I miss the addition of a causal explanation of the results. In simpler terms, why such results were obtained.

To be clear we have highlighted the causal explanation of the results observed in our study. See L193-199; L210-214; L221-228; L231-239; L242-250; L257-262; L265-267; L273-283; L287-297; L304-312.

I suggest supplementing the methodology with a CART analysis.

Thanks for your suggestion. But we are happy with our aims’ achievement into this manuscript. As you already read, we supplemented our analysis with a PCA. This enabled us to show the differences among the studied land uses and the soil layers. By using a CART analysis, we’ll just put here “more of the same thing” by showing clear differences among land uses and soil layers, that we have previously showed with two statistical approaches (univariate and multivariate analyses).

Round 2

Reviewer 2 Report

Thanks to the authors for their revisions, I have no further comments.

Reviewer 3 Report

No modification commentsNo modification comments